

# Classification and prediction of spinal disease based on the SMOTE-RFE-XGBoost model

Biao Zhang[1], Xinyan Dong[2], Yuwei Hu[2], Xuchu Jiang[2] and Gongchi Li[3]

[1] School of Computer Science, Liaocheng University, Liaocheng, Shandong, China
[2] School of Statistics and Mathematics, Zhongnan University of Economics and Law, Wuhan, Hubei, China
[3] Union Hospital Affiliated to Tongji Medical College, Huazhong University of Science and Technology, Wuhan, Hubei, China

## ABSTRACT

Spinal diseases are killers that cause long-term disturbance to people with complex and diverse symptoms and may cause other conditions. At present, the diagnosis and treatment of the main diseases mainly depend on the professional level and clinical experience of doctors, which is a breakthrough problem in the field of medicine. This article proposes the SMOTE-RFE-XGBoost model, which takes the physical angle of human bone as the research index for feature selection and classification model construction to predict spinal diseases. The research process is as follows: two groups of people with normal and abnormal spine conditions are taken as the research objects of this article, and the synthetic minority oversampling technique (SMOTE) algorithm is used to address category imbalance. Three methods, least absolute shrinkage and selection operator (LASSO), tree-based feature selection, and recursive feature elimination (RFE), are used for feature selection. Logistic regression (LR), support vector machine (SVM), parsimonious Bayes, decision tree (DT), random forest (RF), gradient boosting tree (GBT), extreme gradient boosting (XGBoost), and ridge regression models are used to classify the samples, construct single classification models and combine classification models and rank the feature importance. According to the accuracy and mean square error (MSE) values, the SMOTE-RFE-XGBoost combined model has the best classification, with accuracy, MSE and F1 values of 97.56%, 0.1111 and 0.8696, respectively. The importance of four indicators, lumbar slippage, cervical tilt, pelvic radius and pelvic tilt, was higher.

## INTRODUCTION

Spinal diseases are one of the most common diseases in modern society and are also listed as one of the top 10 diseases affecting health rated by the World Health Organization. According to the Guidelines for the Diagnosis, Treatment and Rehabilitation of Cervical Spondylosis 2019, the prevalence of cervical spondylosis in China is approximately 3.8–17.6%, and nearly 150 million people in China suffer from cervical spondylosis. A study published in The Lancet showed that of all disease types, cervical spine pain cost the Chinese people the 9th most healthy life expectancy in 2017, up from the 21st in 1990

Corresponding author
Xuchu Jiang,
xuchujiang@zuel.edu.cn

(*Zhou et al., 2019*). A survey by the Professional Committee for the Prevention and Treatment of Cervical and Low Back Diseases of the China Association for the Promotion of Healthcare International Exchange showed that the prevalence of cervical spondylosis among young adults aged 20–40 years was as high as 59.1% (*Karki et al., 2015*).

Spinal diseases are caused by pathological changes in the bones, intervertebral discs, ligaments, and muscles of the spine, which in turn compress and stimulate the spinal cord, spinal nerves, blood vessels and vegetative nerves, resulting in complex and diverse symptoms (*Byrne, Waxman & Benzel, 2000*) as well as degenerative and infectious diseases of the spine, leading to low back pain and spinal pain or disability and paralysis in more severe cases. At the same time, spinal conditions are also directly or indirectly related to nerve and organ functions in the human body, which can lead to a series of complex diseases. At present, diagnosis and treatment mainly depend on the professionalism and clinical experience of doctors, which places a heavy burden on the problem of low efficiency. Furthermore, the uneven distribution of medical resources in China and the disparity in diagnostic results have made it almost impossible for patients with spinal diseases to seek timely medical care or have accurate diagnosis results, resulting in delayed diagnosis.

In recent years, machine learning and deep learning have played a huge role in many industries. As clinical data always have high dimensionality and large sample sizes, various algorithms can be used as analytical tools to assist in clinical diagnosis. With the advent of the era of healthcare based on big data, artificial intelligence and machine learning technologies can contribute to the rapid and accurate diagnosis of spinal diseases (*D'Angelo et al., 2022*; *Cabitza, Locoro & Banfi, 2018*) and assist physicians in preoperative planning and postoperative outcome prediction, which could help to improve diagnostic efficiency and reduce the burden on medical staff and the rate of misdiagnosis.

## Related works

### An empirical study on spinal problems in primary and middle school students

In recent years, spinal disorders have become increasingly prevalent at a younger age, and many adolescents, even children, suffer from the pain and distress caused by spinal problems. In addition, the prevalence of spinal disorders among adolescents is still increasing due to factors such as incorrect sitting posture, excessive fatigue caused by long-term maintenance of a specific posture, and frequent use of electronic devices.

*Kadhim et al. (2020)* selected a total of 36,728 school students aged 6–18 years in 192 primary and secondary schools in Yuzhong County, Gansu Province, and concluded that the prevalence of spinal abnormalities among primary and secondary school students in Yuzhong County, Gansu Province, was 1.26%. *Li, Zhang & Rong (2021)* monitored 1,902 primary and secondary school students in Lhasa, Shigatse, Nagqu, Chamdo, Shannan city and Ali area and obtained a prevalence of 1.26% for spinal curvature abnormalities in primary and secondary school students. *Qi et al. (2021)* adopted a stratified cluster random sampling method and selected 1,884 primary and secondary school students from grade 4 to grade 3 in Hongkou District for screening of spinal curvature abnormalities and a questionnaire survey of influencing factors. The results showed that the detection rate of

spinal curvature abnormalities in primary and secondary school students in the Hongkou District was 7.2%.

### Research on the application of different learning algorithms in the diagnosis of spinal diseases

A new research direction in the field of artificial intelligence is the cross integration of artificial intelligence technology (machine learning and neural network models) and spinal surgery. At present, many scholars have applied machine learning, deep learning, and computer technology to the diagnosis of spinal diseases and have achieved good results.

In terms of regression analysis methods, *Attiah et al. (2019)* divided 160 study subjects into four age groups, measured a series of spinal sagittal imaging parameters, analyzed data variance and correlation by one-way ANOVA, the least significant difference method, and the Person test, and identified spinal sagittal imaging parameters that were independently correlated with age by linear regression analysis.

In the machine learning algorithm, *Zhu et al. (2022)* used 348 patients with ankylosing spondylitis as the study subjects, used LASSO, random forest and support vector machine recursive feature elimination to screen feature variables and build prediction models, and finally obtained better prediction results with AUCs of 0.878 and 0.823 on the training and validation sets, respectively. *Müller et al. (2022)* used 8,374 patients who had undergone surgery for degenerative disorders of the spine as the research subjects and performed feature selection and model construction by LASSO and ridge regression methods to finally reach the goal of developing a parsimonious model to predict degenerative thoracic, lumbar, or cervical spine. The aim was to develop a parsimonious model to predict multidimensional outcomes in patients undergoing surgery for thoracic, lumbar or cervical degenerative disorders. Based on orthopedic medical data, *Li & Zhang (2020)* constructed and designed a platform for an orthopedic auxiliary diagnosis classification prediction model based on the extreme gradient boosting (XGBoost) algorithm to further realize the auxiliary diagnosis of orthopedic diseases. *Wang et al. (2018)* proposed combining diffusion tensor imaging (DTI) metrics with machine learning algorithms to accurately classify controls and spinal cervical spondylosis (CSCS). The support vector machine (SVM) classifier produced 95.73% accuracy, 93.41% sensitivity, and 98.64% specificity, which showed significant classification performance advantages.

*Shen, Wu & Suk (2017)* worked on high-performance deep learning algorithms for medical image processing and analysis, proposed a novel end-to-end multitask structure correlation learning network (MMCL-Net) for the simultaneous detection, segmentation, and classification of three spinal structures (intervertebral disc, vertebrae and neural foramina), and locally optimized the model to achieve a more stable dynamic equilibrium state. *Hu et al. (2018)* used a long-term short-term memory deep learning network to identify people with chronic low back pain based on human balance and body swing performance in the standing test with an accuracy of 97.2% and a recall rate of 97.2%. *Jamaludin et al. (2017)* used lumbar spine magnetic resonance imaging (MRI) images as input and classified different tasks by a convolutional neural network (CNN) model with a model accuracy up to 95.6%. *Pedersen et al. (2020)* systematically analyzed the accuracy of

different models for predicting the postoperative efficacy of lumbar disc herniation. Compared with the traditional machine learning model, the model based on deep learning can better predict the postoperative efficacy of patients.

For the dataset, *Raihan-Al-Masud & Mondal (2020)* focused on the application of machine learning algorithms for predicting spinal abnormalities. As a data preprocessing step, univariate feature selection as a filter-based feature selection and principal component analysis (PCA) as a feature extraction algorithm are considered. Several machine learning approaches, namely, support vector machine (SVM), logistic regression (LR), and bagging ensemble methods, are considered for the diagnosis of spinal abnormalities.

### The application of ET and RFE in feature selection

In high-dimensional datasets, feature selection is a key step in extracting important features, and it is also the basis for subsequent modeling. Tree-based feature selection and recursive feature elimination (RFE) are common methods for feature selection that are widely used in different fields for different studies and have achieved good results.

In the medical field, *Hu et al. (2021)* used preoperative magnetic resonance images of patients with epithelial ovarian tumors. Image features are extracted from the three-dimensional region of interest manually sketched on the axial T2 weighted imaging (T2WI) image. Four feature selection methods and seven machine learning classifiers are combined in pairs. The results show that the RFE-KNN model combined with RFE and the K-nearest neighbor (KNN) classifier has the best performance. *Zhang et al. (2022)*, based on the physical examination data of a health examination institution in Urumqi in 2018, used three feature selection methods, RFE, measured resting metabolic rate (MRMR) and least absolute shrinkage and selection operator (LASSO), combined with two model explanatory methods of variable importance and linear interpolation with maximum entropy (LIME), to process the metabolic syndrome risk prediction model before and after modeling. *Li & Liu (2020)* used the gene epitope data of spontaneous premature birth (SPB) as a basis, used SVM-RFE for gene feature selection, and compared it with other machine learning and feature selection methods. SPB biomarkers were comprehensively screened out. *Tan et al. (2021)* used logistic regression (LR), SVM-RFE and elastic net methods to preliminarily screen the characteristic variables of the main factors of liver cirrhosis complicated with hepatic encephalopathy based on the data of patients with liver cirrhosis with complete medical records. LR and multilayer perceptron (MP) were used as meta-learners to construct a stacked generalization (stacking) heterogeneous integrated classification model. *Gitto et al. (2022)* studied 158 patients with chondroosseous tumors as research objects. This article classified and diagnosed atypical cartilaginous tumors (ACTs) and grade II chondrosarcomas of long bones (CLB2) using machine learning based on magnetic resonance imaging (MRI) radiomics, applied LASSO and RFE to determine the feature sets for model training, and balanced the datasets using a synthetic minority oversampling technique (SMOTE). The final model achieved good results with 98% and 80% accuracy in ACT and CLB2, respectively.

*Cao et al. (2021)* extracted the 10 most helpful features for predicting purchase behavior based on the ET algorithm and used logistic regression and the support vector machine algorithm to construct a purchase behavior prediction model. Finally, the above two algorithms were fused based on the method of soft voting and obtained a good prediction effect. *Kurniawan et al. (2022)* used Harris hawks optimization (HHO) and support vector regression (HHO-SVR) to build a prediction model for ozone concentration in 14 partitions of JABODETABEK. Recursive feature elimination and support vector regression (RFE-SVR) were used to select the important predictors, the HHO-SVR method and support vector regression (SVR) were used to establish the prediction model, the HHO algorithm was used to optimize the values of their parameters, and the final HHO-SVR model obtained a better conclusion. *Wang, Yang & Dai (2021)* took dam displacement as the research object and safety monitoring data as the research basis and proposed a prediction model for the dam displacement of tailings dams based on recursive feature elimination, random forest and limit gradient enhancement and compared it with prediction models such as XGBoost, LSTM neural networks, BP neural networks and SVR. The results showed that the RFE-RF-XGBoost model had an average relative error of 3.93%, which was lower than that of the XGBoost model.

This section first describes the results of an empirical investigation of spine problems in Chinese primary and secondary school students. The results of different algorithms, such as regression analysis, machine learning algorithms, and deep learning algorithms related to the diagnosis of spinal disorders, are also presented separately. Since ET and RFE are often used when dealing with high-level datasets and extracting their important features, the first part also presents the application of the combined model derived from ET and RFE for feature selection.

## Research framework

The physical angle index of human bone can comprehensively reflect the health status of the spine. A small change in a certain index may cause complex chain reactions of other indicators. This study aims to use physical indicators that are relatively unaffected by the cascade effect and then perform feature selection to make the selected features more representative. Furthermore, a machine learning classifier is used to develop a feature selection-classification combination model, and the best performing model is comprehensively compared for spinal health diagnosis. Finally, there is a category imbalance problem in the original sample used in the article, which is the key to the superiority of feature selection and model construction results. In the data processing stage, dealing with the category imbalance by the SMOTE algorithm is one of the focus parts of this article. The research framework of this article is shown in Fig. 1.

## RESEARCH METHOD

According to Fig. 1, the article conducts the empirical analysis part with three main sections: data preprocessing, feature selection and model construction. The important step in the data preprocessing section is the category imbalance treatment using the SMOTE algorithm; the feature selection section uses three models, LASSO, ET and RFE, and the

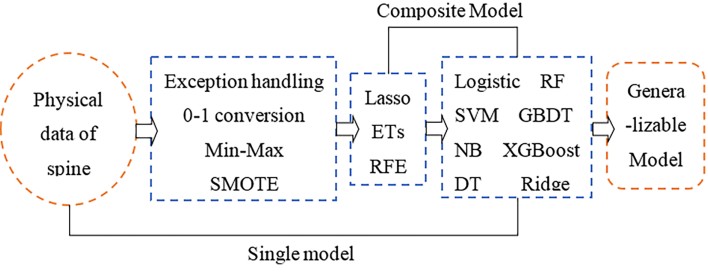

**Figure 1 Research framework.**

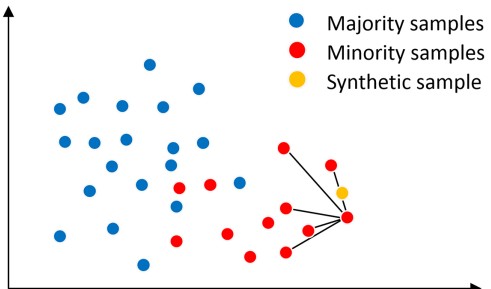

**Figure 2 SMOTE algorithm mechanism.**

model construction section uses eight different models. To highlight the significant part of the article, only the SMOTE algorithm, XGBoost model, ET model, and RFE model are described in detail below.

## Principle of the SMOTE algorithm

The category imbalance problem refers to the situation where the number of training samples of different categories in a classification task varies greatly. *Chawla et al. (2002)* proposed the SMOTE algorithm in 2002, which is based on the principle of filtering samples $x_i$ and $x_j$ and the corresponding random numbers $0 < \lambda < 1$ from a small set of samples, constructing new samples $x_n = x_i + \lambda(x_j - x_i)$ by the relationship between the two samples, which means that the minority samples are analyzed and new samples are added to the dataset by artificial synthesis based on the minority samples. The SMOTE algorithm is essentially an oversampling; it does not sample on the data space but in the feature space, so its accuracy will be higher than the traditional sampling method.

In the SMOTE algorithm, the neighborhood space is determined by the K-nearest neighbor method, as shown in Fig. 2, assuming that the main sample x has K = 5 nearest neighbor samples on the feature space, from which a nearest neighbor sample x' is randomly selected, and then a new sample $x^{new}$ is generated at a random position on the connection line between the main sample x and the nearest neighbor sample x'. The synthesis of the new sample $x^{new}$ is calculated as shown in Eq. (1) is shown. where x is the selected master sample, x' is the randomly selected nearest neighbor sample of x, $x^{new}$ is the synthesized new sample, and rand(0, 1) denotes the random number on the generated (0, 1).

$$x^{new} = x + rand(0, 1) \times (x' - x) \tag{1}$$

---

**Algorithm 1  SMOTE**

**Input:** Training set $S = \{(x_i, y_i), i = 1, 2, \ldots, N, y_i \in \{+, -\}\}$; Sample size of majority classes $N^-$, Sample size of minority classes $N^+$,

$N^+ + N^- = N$; Imbalance ratio $IR = \dfrac{N^-}{N^+}$; Sample rate $SR$; Proximity parameter $K$.

**Output:** The training set after oversampling $S = \{(x_i, y_i), i = 1, 2, \ldots, N + N^+ \times SR, y_i \in \{+, -\}\}$

1:  Remove all majority class and minority class samples from the training set $S$ to form a majority class training sample set $S^-$ and minority classes of training sample sets $S^+$.

2:  Set the newly generated sample set $S^{New}$ to empty

3:  *for* $i = 1 : N^+ \times SR$

4:  Randomly choose a number from $[1, \ N^+]$, find the corresponding sample $x$ in $S^+$.

5:  find the $K$ neighbor sample of the main sample $x$ from $S^+$, and set it into the neighbor sample group $S^{Near}$.

6:  Randomly choose a number from $[1, K]$, find the corresponding main neighbor sample $x'$ from $S^{New}$.

7:  Calculate the new minority class sample: $x^{new} = x + rand \times (x' - x)$, in which $rand \in [0, 1]$.

8:  Add $x^{new}$ to $S^{New}$: $S^{New} = S^{New} \cup x^{new}$

9:  Set the neighbor sample group $S^{Near}$

10:  *end for*

return training set after oversampling $S' = S^- \cup S^{New}$

---

The pseudocode of Algorithm 1 of the SMOTE algorithm is shown:

## The XGBoost principle

The XGBoost algorithm belongs to the time-series supervised learning model, which is an optimization and enhancement of the gradient boosting tree, by combining several learners to produce a boosting algorithm with strong learners. The loss function of XGBoost includes its own loss and regularization parts, and the addition of the second-order Taylor expansion of the error part and the $L_1$ and $L_2$ regularization terms makes it more accurate and has stronger generalization ability. The loss function of XGBoost is shown in Eqs. (2) and (3).

$$L_t = \sum_{i=1}^{J} \left[ G_{tj}\omega_{tj} + \frac{1}{2}\left(H_{tj} + \lambda\right)\omega_{tj}^2 \right] + \gamma J \tag{2}$$

$$G_t j = \sum_{x_i \in R_{tj}} g_{ti}, \qquad H_t j = \sum_{x_i \in R_{tj}} h_{ti} \tag{3}$$

where $L_t$, which is denoted as the loss function of XGBoost, $x_i$ is the input samples, and $G_{tj}$ and $H_{tj}$ denote the sum of the first-order derivatives and second-order derivatives of all input samples for the $t$th decision tree mapping to the leaf node $j$, respectively. $J$ is the number of leaf nodes, $\gamma$ indicates the difficulty of the node cut, $\lambda$ is the regularization factor of $L_2$, $g_{ti}$ and $h_{ti}$ are the first-order and second-order derivatives of the $i^{th}$ sample at the $t^{th}$ weak learner, respectively, and $\omega_{tj}$ is the optimal value of the $j^{th}$ leaf node. Finally, the loss function is minimized to obtain the optimal solution $\omega_{tj}$ for all J leaf node regions and each leaf node region for the $t^{th}$ decision tree optimum.
**Table 1 Evaluation index.**

| Indicators | Meaning |
|---|---|
| Accuracy | The proportion of the number of paired samples to the total number of samples |
| Recall rate | The proportion of the number correctly divided into positive cases to all positive cases |
| F1 value | Comprehensive index of recall and precision, measuring the equilibrium point on the curve $P\_R$ |
| AUC value | The area enclosed by ROC curve and horizontal axis is used to measure the generalization performance of the model |

## Principle of tree-based feature selection

According to the random forest algorithm, the importance of each attribute can be calculated through tree model training, and the value of importance can help us select important features. This study uses the ET algorithm, which directly uses random features and random threshold division, and the shape and difference of each decision tree will be increasingly large and random.

## Principle of recursive feature elimination (RFE)

The recursive elimination feature method uses a base model for multiple rounds of training. After each round of training, the features of several weight coefficients are eliminated, and then the next round of training is carried out based on the new feature set. It uses the model accuracy to identify which attributes (and attribute combinations) contribute the most to the prediction of target attributes.

## Model evaluation metrics

In this article, the accuracy, recall, F1 value, area under the curve (AUC) value and mean square error (MSE) were used to evaluate the performance of different models. MSE is used to measure the difference between the predicted value and the real value of the regression task, which is the mean of the square sum of the error between the predicted value and the real value, which is calculated as Eq. (4).

$$MSE = \frac{1}{n} \sum_{i=1}^{m} \omega_i (y_i - \widehat{y}_i)^2 \tag{4}$$

where $y_i$ is the real value, $\widehat{y}_i$ is the predicted value, $\omega_i > 0$, and $n$ is the number of samples. The smaller the MSE is, the smaller the difference between the predicted value and the real value. Therefore, the smaller the MSE is, the better the model.

Other indicators are calculated based on the confusion matrix. The confusion matrix is a standard format that represents the accuracy evaluation. Each column represents the prediction category, and each row represents the true attribution category of the data. According to the confusion matrix, various indicators can be calculated. The evaluation index and its meaning are shown in Table 1, and the higher the value is, the better the model.

This section revolves around the theoretical structure of the research approach of this article, introducing for the first time a research framework for spinal health diagnosis by feature selection combined with a physical indicator model selected by machine learning.

**Table 2 Characteristic description.**

| Variables | Feature name | Feature symbol | Feature Type |
|---|---|---|---|
| Independent variable | Pelvic incident | $x_1$ | Continuous type |
| | Pelvic tilt | $x_2$ | |
| | Lumbar lordosis angle | $x_3$ | |
| | Sacral slope | $x_4$ | |
| | Pelvic radius | $x_5$ | |
| | Degree spondylolisthesis | $x_6$ | |
| | Pelvic slope | $x_7$ | |
| | Direct tilt | $x_8$ | |
| | Thoracic slope | $x_9$ | |
| | Cervical tilt | $x_{10}$ | |
| | Sacrum angle | $x_{11}$ | |
| | Scoliosis slope | $x_{12}$ | |
| Dependent variable | class_att | $y$ | Discrete type |

The principles of the SMOTE algorithm, XGBoost, and tree-based feature selection and RFE algorithms employed in the model are presented. In addition, a total of five evaluation metrics, accuracy, recall, F1 value, AUC value, and mean square error (MSE), were adopted as the evaluation metrics of the model.

## EMPIRICAL ANALYSIS

### Data sources

In this article, the dataset is from the exploration case of dichotomous classification of back pain symptoms on the Kaggle website, which is a dichotomous classification problem to detect whether the spine of a person is healthy by collecting physical data from the human spine and pelvis and other parts. The dataset has 310 sets of sample observations and 13 attributes, of which 12 attributes are numerical variables and serve as independent variables and one attribute is a categorical variable and serves as the dependent variable, with the characteristics described in Table 2.

### Data preprocessing

#### Abnormal value processing

According to the box diagram, histogram, density curve, *etc.*, there are extreme points in the data, among which $x_1$ (incidence of pelvic fractures), $x_3$ (lumbar lordosis angle), $x_4$ (sacral inclination angle) and $x_6$ (lumbar spondylolisthesis) have more obvious extreme points. Based on the results of basic descriptive statistical analysis, the maximum incidence of pelvic fractures is 129.83% > 100%, which is not consistent with the actual situation; there is a large gap between the maximum values of the lumbar kyphosis angle and sacral inclination angle and their average level and 75% quantile; the maximum value of lumbar spondylolisthesis is obviously separated from the overall level, which can be judged as an

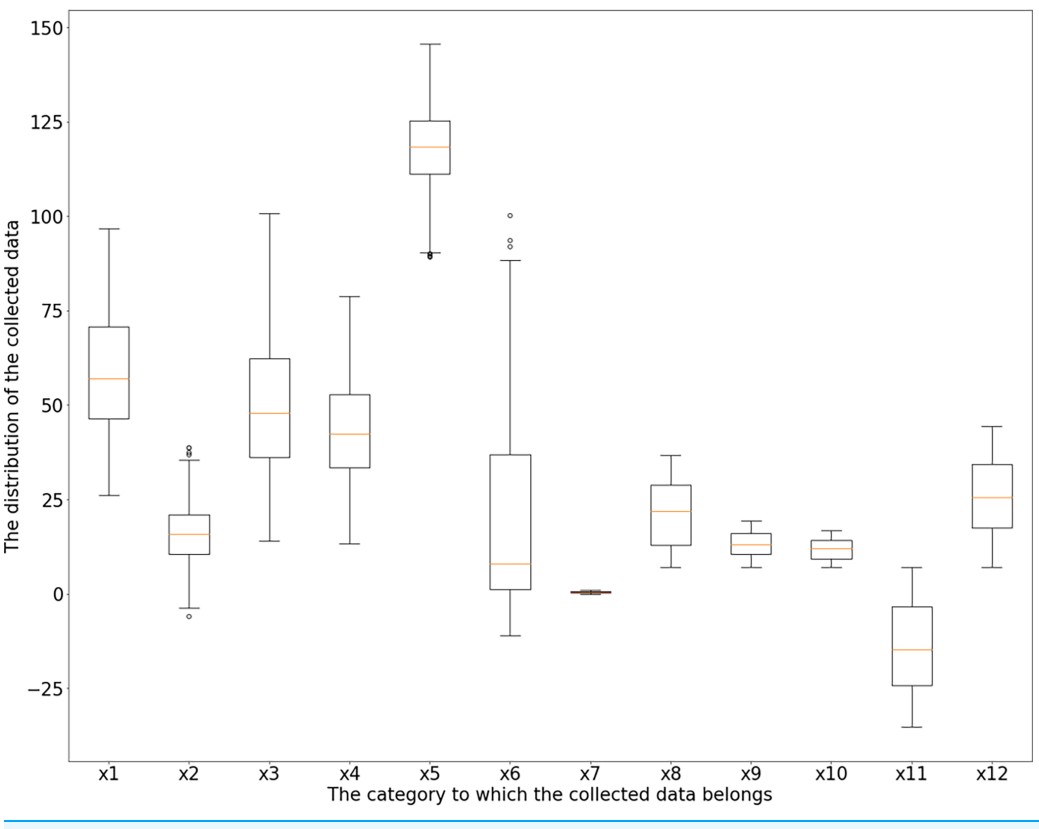

**Figure 3 Feature boxplot.**

abnormal value. Taking the upper and lower boundaries of the box diagram as the critical point of the abnormal value, the abnormal values of the six characteristics of $x_1$ (incidence of pelvic fractures), $x_2$ (pelvic tilt), $x_3$ (lumbar lordosis angle), $x_4$ (sacral inclination angle), $x_5$ (pelvic radius) and $x_6$ (lumbar spondylolisthesis) are identified and eliminated. Among the above six features, the data values larger than the upper edge and smaller than the lower edge of the box line graph are considered outliers, and all samples corresponding to the outliers are considered abnormal samples and are rejected. According to the results, the abnormal values of the above six features are 3, 13, 1, 1, 11 and 10. The new dataset after rejection contained 279 samples, and the box diagram and histogram of the new dataset are shown in Figs. 3 and 4.

In Fig. 3, the horizontal axis is the category to which the collected data belong, and $x_1 \sim x_{12}$ refers to the feature symbol in Table 2. The vertical axis is the distribution of the collected data, and 25 is an interval.

In Fig. 4, the horizontal axis is the distribution of the collected data, with 20 as an interval. The vertical axis is the density distribution of the collected data, and the orange line is the density curve of the collected data.

### Category 0–1 conversion

Because the variable is nonnumerical, it is converted to a 0–1 variable. Among them, the normal spine is marked as 0, and the abnormal spine is marked as 1.

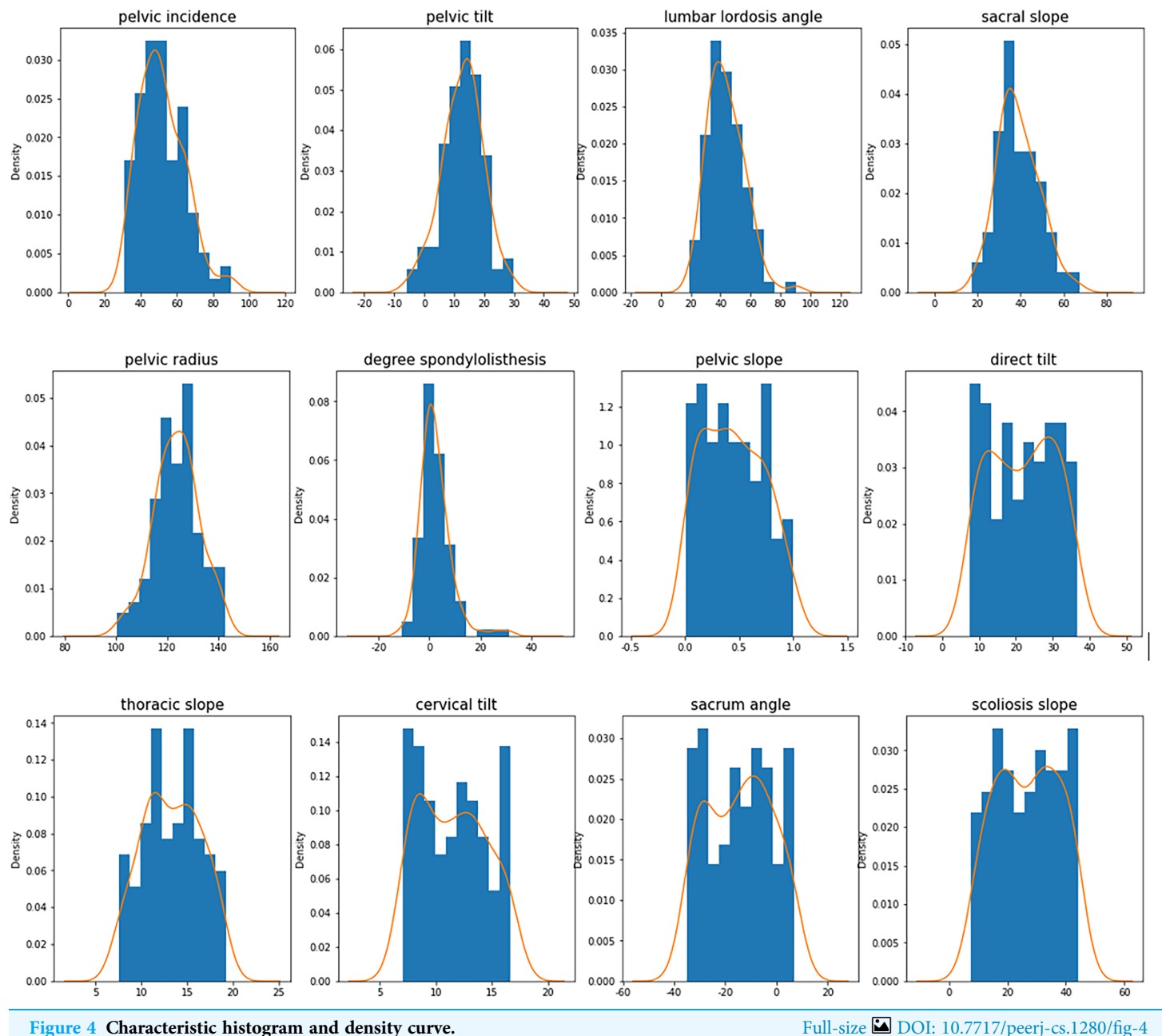

**Figure 4 Characteristic histogram and density curve.**

### Min-max standardized processing

The dataset was divided into two parts: independent variables and labels. Data standardization (Min-Max) was applied to the independent variables to restrict the data range to [0, 1], which eliminated the errors caused by the different magnitudes and transformed all the data into positive values. The *standardization* is used as Eq. (5).

$$x' = \frac{x - \min(x)}{\max(x) - \min(x)} \tag{5}$$

**Table 3  Feature importance matrix.**

| Variables | LASSO | ETs | RFE |
|---|---|---|---|
| $x_1$ | 0.0000 | 0.0762 | 1 |
| $x_2$ | 0.2166 | 0.0746 | 1 |
| $x_3$ | −0.1679 | 0.0733 | 1 |
| $x_4$ | −0.5076 | 0.0785 | 1 |
| $x_5$ | −0.8561 | 0.1297 | 1 |
| $x_6$ | 1.5406 | 0.2854 | 1 |
| $x_7$ | 0.0070 | 0.0409 | 6 |
| $x_8$ | 0.0024 | 0.0448 | 5 |
| $x_9$ | −0.0526 | 0.0514 | 2 |
| $x_{10}$ | 0.0619 | 0.0520 | 1 |
| $x_{11}$ | −0.0602 | 0.0490 | 4 |
| $x_{12}$ | −0.0924 | 0.0441 | 3 |

where $x$ is the original value of a feature, $min(x)$ is the minimum of $x$, $\max(x)$ is the maximum of $x$ and $x'$ is the normalized value of $x$.

### *Class imbalance processing based on SMOTE*

After excluding the abnormal values, the dataset showed 99 samples with normal spines and 180 samples with abnormal spines, with a large difference in the number of 0–1 samples. Therefore, the SMOTE algorithm was used to process the samples with class imbalance. Specifically, the principle is for a minority class sample. Using the K nearest neighbor method (the k value needs to be specified in advance), the distance xi is obtained. The nearest k minority samples, According to the principle of the SMOTE algorithm, the process of synthesizing a new sample will randomly select one of the five nearest neighbor samples, multiply the Euclidean distance of the two samples by a random number between (0, 1), and determine the exact location of the synthesized sample based on the new distance. For this dataset, the SMOTE function in the imblearn library (a Python library for handling unbalanced data) is called for oversampling, and the default value of 5 is used for the k_neighbors parameter (the number of neighboring samples k). In the original data, normal spines are the minority sample, and abnormal spines are the majority sample. The SMOTE oversampling process was performed for the minority class samples, and no operation was performed for the majority class samples, which finally expanded the minority class samples to 180 cases. The same number of cases as the majority class samples, of which 99 cases were original samples and 81 cases were synthetic samples, were integrated to form the normal spine class samples.

### Feature selection

The features are selected and compressed using three methods: LASSO, ET, and RFE. All three methods are direct selections of the original features without any linear combination or transformation, and the selected features are consistent with the original features. The feature importance matrix is shown in Table 3.

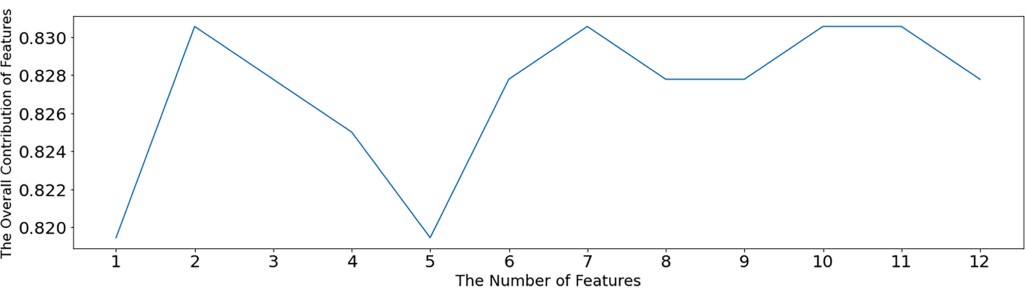

**Figure 5 Optimal feature number selection under SVM-RFE.**

### LASSO compression

When performing LASSO compression, the LASSOCV function is used in this article, and after debugging the optimal regularization parameter alpha, the optimal regularization parameter value of 0.001 is selected, and the final output is the mean value of the residuals and the degree of feature importance using 10-fold cross-validation. According to the LASSO compression results, only features with positive values of the weight parameter are selected; that is, features with 0 and negative values of the parameter are excluded. The final feature selection results are $x_2$ (pelvic tilt), $x_6$ (lumbar slippage), $x_7$ (pelvic obliquity), $x_8$ (direct tilt), and $x_{10}$ (neck tilt).

### ET method

The importance values of the features under the ET method performed normally. According to the results, six features with an importance value higher than 0.06 are selected: $x_1$ (pelvic fracture incidence), $x_2$ (pelvic tilt), $x_3$ (lumbar anterior convexity angle), $x_4$ (sacral tilt angle), $x_5$ (pelvic radius), and v (lumbar slip degree).

### RFE method

The SVM-RFE algorithm is used to test the optimal number of features, as shown in Fig. 5. When the number of features is 2, 7, 10 and 11, the overall contribution of features reaches the maximum, which is 0.83056. When the number of features is 2, a sparse solution is obtained, and this solution is not desirable. Therefore, the minimum number of features except 2, namely, seven features, is selected. After setting the number parameter to 7, the SVM-RFE algorithm is still used for feature selection, and the value of 1 in Table 3 is the selected feature. The other non1 values are the unselected features, which are $x_1$ (pelvic fracture incidence), $x_2$ (pelvic tilt), $x_3$ (lumbar anterior convexity angle), $x_4$ (sacral tilt angle), $x_5$ (pelvic radius), $x_6$ (lumbar slippage), and $x_{10}$ (cervical tilt).

In Fig. 5, the horizontal coordinate indicates the number of retained features, and the vertical coordinate indicates the overall contribution of the features. Using the average of the SVM model accuracy to represent the contribution level, the higher the accuracy, the greater the overall contribution of the features.

According to the above results and analysis, there are three feasible results for feature selection: LASSO compresses the features to five, selects six features based on tree-based feature selection, and selects seven features based on recursive feature elimination. These

three feature selection methods and the selection results are used to construct different classification models.

## Model construction and algorithm evaluation

Model construction is divided into two parts. The first part constructs classification models using original features without feature selection and constructs logistic regression, SVM, NB, decision tree (DT) random forest (RF), gradient boosted tree (GBT), XGBoost, and ridge regression models, with a total of eight classification models. The second part is to build the combination model of feature selection-classification; that is, the three methods of feature selection are combined with the seven classification models except the ridge regression model, and 21 different combination models are obtained. In the above two parts, 29 different classification models and combination models were obtained. Five evaluation indexes were compared: accuracy, recall, F1 value, AUC value and MSE. The results of the evaluation metrics are shown in Table 4.

In the single-model part, there are eight models, and the precision of all eight models is greater than 0.8, MSE is less than 0.3, recall, F1 value, and AUC are slightly inferior, but the overall level of the models is good, and the differences between models are not obvious. Among them, the accuracies of SVM, RF, XGBoost, and ridge regression are all greater than 0.9, and the accuracy of ridge regression is the highest at 0.9286, while the AUC and MSE values of RF and XGBoost reach the maximum and minimum at 0.8623 and 0.1389, respectively.

In the combined model section, there are 21 models, and the overall model performance under different feature selection methods differed to a lesser extent. However, there are significant differences between individual model performances. Under the LASSO method, the overall accuracy of the models is slightly better than that of the single model, and the recall, F1 value, AUC value, and MSE value are inferior to those of the single model. The accuracy of LASSO-DT is the lowest at 0.7955, and the accuracy of the remaining six models is higher than 0.87, among which LASSO-SVM and LASSONB have the highest accuracy at 0.9444. Under the ET method, the accuracy of the models is higher than 0.86. The model accuracy is greater than 0.86, in which ET-XGBoost has the highest accuracy of 0.9744, and its MSE value and ET-RF both reach the minimum of 0.1296. Under the RFE method, the overall performance of the model is better, and the best performance of the other four indexes except the recall is in this part. The model accuracy is greater than 0.85, in which RFE-XGBoost has the highest accuracy. The accuracy of RFE-XGBoost is the highest, reaching 0.9756, and its MSE value and ET-SVM are both the smallest, at 0.1111.

In summary, the accuracy of the SMOTE-RFE-XGBoost model is the highest in all models (97.56%), the MSE value is the lowest in all models (0.1111), the AUC value is the second highest in all models (0.8834), it is only slightly lower than the top in all models (0.8844), and the F1 value is the third highest in all models (0.8696). Thus, the optimal model of this study is the SMOTE-RFE-XGBoost model.

**Table 4  Results of model evaluation indexes.**

| Type | Models | Accuracy | Recall rate | F1 value | AUC value | MSE value |
|---|---|---|---|---|---|---|
| Single model | Logistic | 0.8605 | 0.6727 | 0.7551 | 0.7798 | 0.2222 |
| | SVM | 0.9111 | 0.7455 | 0.8200 | 0.8350 | 0.1667 |
| | NB | 0.8222 | 0.6727 | 0.7400 | 0.7609 | 0.2407 |
| | DT | 0.8519 | 0.8364 | 0.8440 | 0.8427 | 0.1574 |
| | RF | 0.9000 | 0.8182 | 0.8571 | 0.8619 | 0.1389 |
| | GBT | 0.8980 | 0.8000 | 0.8462 | 0.8528 | 0.1481 |
| | XGBoost | 0.9167 | 0.8000 | 0.8544 | 0.8623 | 0.1389 |
| | RR | 0.9286 | 0.7091 | 0.8041 | 0.8262 | 0.1759 |
| Combi-nation model | LASSO-Logistic | 0.9231 | 0.7059 | 0.8000 | 0.8266 | 0.1667 |
| | LASSO-SVM | 0.9444 | 0.6667 | 0.7816 | 0.8158 | 0.1759 |
| | LASSO-NB | 0.9444 | 0.6667 | 0.7816 | 0.8158 | 0.1759 |
| | LASSO-DT | 0.7955 | 0.6863 | 0.7368 | 0.7642 | 0.2315 |
| | LASSO-RF | 0.8780 | 0.7059 | 0.7826 | 0.8091 | 0.1852 |
| | LASSO-GBT | 0.8780 | 0.7059 | 0.7826 | 0.8091 | 0.1852 |
| | LASSO-XGBoost | 0.9250 | 0.7255 | 0.8132 | 0.8364 | 0.1574 |
| | ETs-Logistic | 0.8864 | 0.7647 | 0.8211 | 0.8385 | 0.1574 |
| | ETs-SVM | 0.9091 | 0.7843 | 0.8421 | 0.8571 | 0.1389 |
| | ETs-NB | 0.8780 | 0.7059 | 0.7826 | 0.8091 | 0.1852 |
| | ETs-DT | 0.8723 | 0.8039 | 0.8367 | 0.8493 | 0.1481 |
| | ETs-RF | 0.9111 | 0.8039 | 0.8542 | 0.8669 | 0.1296 |
| | ETs-GBT | 0.8667 | 0.7647 | 0.8125 | 0.8297 | 0.1667 |
| | ETs-XGBoost | 0.9744 | 0.7451 | 0.8444 | 0.8638 | 0.1296 |
| | RFE-Logistic | 0.8864 | 0.7647 | 0.8211 | 0.8385 | 0.1574 |
| | RFE-SVM | 0.9535 | 0.8039 | 0.8723 | 0.8844 | 0.1111 |
| | RFE-NB | 0.8750 | 0.6863 | 0.7692 | 0.7993 | 0.1944 |
| | RFE-DT | 0.8542 | 0.8039 | 0.8283 | 0.8406 | 0.1574 |
| | RFE-RF | 0.9302 | 0.7843 | 0.8511 | 0.8658 | 0.1296 |
| | RFE-GBT | 0.8864 | 0.7647 | 0.8211 | 0.8385 | 0.1574 |
| | RFE-XGBoost | 0.9756 | 0.7843 | 0.8696 | 0.8834 | 0.1111 |

## Feature importance ranking

According to the obtained optimal model SMOTE-RFE-XGBoost, the importance ranking of the features is performed by applying the model, and the features involved in the ranking are processed by feature selection instead of importance analysis for all features. The results of the importance analysis are shown in Fig. 6. According to the results, the importance of lumbar slippage was the highest, and the importance of sacral tilt angle, pelvic radius, and lumbar anterior convexity angle were the next highest, while the importance of lumbar slippage was much higher than that of other features. Therefore, this index of lumbar slippage needs extra attention during clinical diagnosis and model construction.

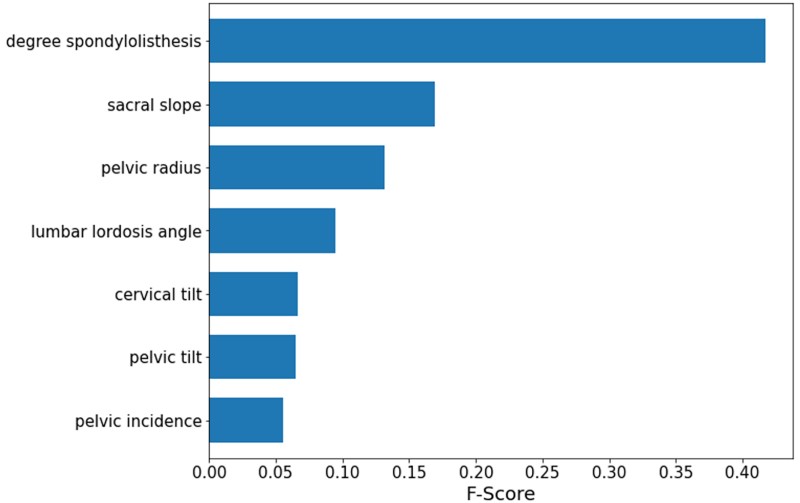

**Figure 6 SMOTE-RFE-XGBoost feature importance ranking.**

This section revolves around an empirical analysis to detect whether the spine of a person is healthy by using physical data collected by the kaggle website on the human spine and pelvis, among other areas. The dataset contains 310 sets of sample observations and 13 attributes. After preprocessing the data, the data were divided into training and testing sets. The features were selected and compressed using three methods: LASSO, tree-based feature selection, and RFE. The first part constructs a classification model using the original features, and the second part constructs a combined feature selection-classification model. Finally, by comparing the accuracy of each model, the optimal model for this study is obtained: the SMOTE-RFE-XGBoost model.

## CONCLUSION AND FUTURE WORK

This article deeply explored category imbalance processing, feature selection, and machine learning model combinations to construct a more quasilateral diagnostic model for spinal diseases. For category imbalance processing, the SMOTE algorithm proposed by *Chawla et al. (2002)* is used for oversampling in this article. Since the dataset used in this article is a public dataset, *Raihan-Al-Masud & Mondal (2020)* used this dataset for machine learning classification exploration, including basic data preprocessing, dataset expansion, feature selection, and model construction. Among them, the data preprocessing did not involve category inequality processing, and none of the final obtained accuracies reached 90%. In contrast, the accuracy of the optimal model finally obtained in this article reached 97.56%, and the classification accuracy was significantly improved. Therefore, the SMOTE category imbalance treatment has an extremely critical impact on model construction, model performance and prediction capability. In terms of feature selection, this article used the Lasso, ETs and RFE algorithms to directly screen features and did not use PCA for linear transformation, which to a certain extent reduced the computational complexity of the model and retained the most original information of the data. In terms of model construction, all feature selection algorithms are modeled in this article to select the best

feature selection scheme; at the same time, models without feature selection are set up for control to explore the differences in the effects produced by feature selection. From Table 4, we can see that the models with feature selection are better than those without feature selection, and then we compare the classification models constructed by different feature selection algorithms horizontally.

The results of this article show that the combined SMOTE-RFE-XGBoost model has the best classification prediction with an accuracy of 97.56%, which improves the accuracy by 5.89% and reduces the MSE value by 20% compared to the XGBoost single model. The SMOTE category imbalance processing algorithm and RFE feature selection improve the model accuracy to some extent, so the performance of the model is significantly improved when combined with the appropriate classification model. In practical applications, the SMOTE-RFE-XGBoost model can improve the accuracy and reliability of spinal disease diagnosis results to a certain extent and can be used as an aid for the clinical diagnosis of spinal diseases. The dataset used in this study is small in both feature dimensions and sample size. Especially in the era of big data healthcare, where there are extremely high-dimensional features and large sample sizes, machine learning and deep learning can better reflect their strong learning ability in big data. Therefore, more studies and surveys are needed, and more data need to be collected for a deeper study of spinal disorders. Finally, this article is dedicated to machine learning algorithms, while deep learning and artificial intelligence have shown more complex learning mechanisms and better learning capabilities in recent years. It is worthwhile to use relevant algorithms to explore spinal disorders more deeply.

This article studied the physical data of the human spine and pelvis and finally obtained a unimodal prediction model with good results. In fact, the symptoms of spinal diseases are complex and diverse, and it is obvious that the unimodal model lacks conviction to diagnose only from one aspect of the clinical data, and the model is not robust, so the prediction results could be easily affected, even leading to incorrect diagnosis. The bootstrap method *Raihan-Al-Masud & Mondal (2020)* used to expand the dataset is our main research direction in the future. Therefore, first, the original data need to be expanded in terms of feature dimension and sample size, and second, the model needs to be expanded into a multimodal feature model (collect multifaceted and multiform clinical information such as textual information, digital information and image information) and build a more accurate diagnostic model through multimodal fusion technology, which could ensure accurate prediction and make the model more stable and scientific. It helps to make faster and better decisions to assist clinical diagnosis.

### Funding
The research is supported by Natural Science Foundations of Shandong Province (Grant No. ZR2021QF036), and by "Guangyue Young Scholar Innovation Team" of Liaocheng University (Grant No. LCUGYTD2022-03). The funders had no role in study design, data collection and analysis, decision to publish, or preparation of the manuscript.

## Grant Disclosures

The following grant information was disclosed by the authors:
Natural Science Foundations of Shandong Province: ZR2021QF036.
"Guangyue Young Scholar Innovation Team" of Liaocheng University:
LCUGYTD2022-03.

## Competing Interests

The authors declare that they have no competing interests.

## Author Contributions

- Biao Zhang conceived and designed the experiments, analyzed the data, performed the computation work, prepared figures and/or tables, authored or reviewed drafts of the article, and approved the final draft.
- Xinyan Dong performed the experiments, performed the computation work, prepared figures and/or tables, authored or reviewed drafts of the article, and approved the final draft.
- Yuwei Hu analyzed the data, performed the computation work, prepared figures and/or tables, authored or reviewed drafts of the article, and approved the final draft.
- Xuchu Jiang conceived and designed the experiments, analyzed the data, performed the computation work, prepared figures and/or tables, authored or reviewed drafts of the article, and approved the final draft.
- Gongchi Li performed the experiments, performed the computation work, prepared figures and/or tables, authored or reviewed drafts of the article, and approved the final draft.

## Data Availability

The raw measurements are available in the Supplemental Files.

## Supplemental Information

Supplemental information for this article can be found online at http://dx.doi.org/10.7717/peerj-cs.1280#supplemental-information.

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
