# Peer review of "Classification and prediction of spinal disease based on the SMOTE-RFE-XGBoost model"

_PeerJ Computer Science, doi:10.7717/peerj-cs.1280_

## Round 0.1 · original submission · Major Revisions

The paper has major issues that need to be addressed to improve its standard.

Reviewer 1 ·

Basic reporting

The writing style requires improvement, for example
1- line 50-51
2- error in line 126
3- The following tile is ambiguous of line 172 " Model Introduction" It would be proposed framework brief description and to contains a brief about the work.
4 the related work part missed the results as of 4,5 and it also need to be more recent
5 reference are not of the same style (4,5,6,,.......)

Experimental design

The data set is public and other researchers applies the same classifiers , however there is no reference to them nor comparison
like this one
Raihan-Al-Masud, M., & Mondal, M. R. H. (2020). Data-driven diagnosis of spinal abnormalities using feature selection and machine learning algorithms. Plos one, 15(2), e0228422.

Validity of the findings

The data imbalance is a huge problem but there was no details about how the oversampling was applied

Reviewer 2 ·

Basic reporting

-The acronyms are incorrect. The correct form is with the initial capital letters representing acronyms such as Recursive Feature Elimination (RFE). This error must be corrected throughout the document.
-The authors must correctly use the verb tenses depending on the Sections.
-Authors must add a Related Works Section.
-After each Section or Subsection must have a brief introduction.
-The English level of the manuscript must be improved.
-Delete the link from line 229.
-Where did the authors obtain the data?
-Figures 2 and 3 show no legends, and the axes cannot be seen. Also, there are no corresponding units on the axes.
-Authors must avoid using Phrasal Verbs in a scientific article.
-After page 23, the Figures and Tables must be eliminated because they are in the manuscript in their respective position.
-The conclusions must be improved, and add future work.

Experimental design

-There is no scenario where they have verified the proposal presented by the authors.
-The experiments they have developed lack more details if they used a simulator or computer program to verify what was obtained.

Validity of the findings

-The authors do not present where they obtained the data.
-The conclusions must be improved, and add future work.

Additional comments

-The acronyms are incorrect. The correct form is with the initial capital letters representing acronyms such as Recursive Feature Elimination (RFE). This error must be corrected throughout the document.
-The authors must correctly use the verb tenses depending on the Sections.
-Authors must add a Related Works Section.
-After each Section or Subsection must have a brief introduction.
-Equations are not numbered, and it is not known which Equation is being cited in the manuscript.
-The English level of the manuscript must be improved.
-Delete the link from line 229.
-Where did the authors obtain the data?
-Figures 2 and 3 show no legends, and the axes cannot be seen. Also, there are no corresponding units on the axes.
-Authors must avoid using Phrasal Verbs in a scientific article.
-The same type of letters must be used in all the Equations.
-The equation found in line 264 is not correctly written.
-After page 23, the Figures and Tables must be eliminated because they are in the manuscript in their respective position.
-The conclusions must be improved, and add future work.

Reviewer 3 ·

Basic reporting

See below

Experimental design

See below

Validity of the findings

See below

Additional comments

Authors have addressed a very good area of research; however, I have some concerns regarding this work.
1. Write up needs proof reading. Thoroughly revise the whole paper so that reviewers would be able to conclude that what you have done.
2. Please, add references in the introduction.
3. In section 3.2.5, the authors dividing the dataset. Please, specify the bases for dividing the data 70:30 ratio and justify the use of 310 data size. It is comparatively very smaller in size.
4. Explain the treatment in SMOTENNN, Kindly, mention the SMOTENN references.
5. In this paper, authors are claiming that there is a proposed model, however, the proposed model is not discussed. Please explained the proposed model in the paper.
6. How you select the best model for implementation. Please justify
7. No mathematical details about the proposed model. Kindly mention it and also properly cite the mathematical equations which the authors are using in the paper.
8. Due to small size of dataset, the results are not justified. (it’s a suggestion to apply K-fold validation (at least 5 folds)) and then mention the average results.
9. There is no benchmark comparison with various model with proposed model
10. Discuss the model selection criteria which is using in a paper?
11. Results are showing on the basis of testing data: however, it is not generalizable. Specified it
12. The code is not showing the same results which has been mentioned and the authors used built-in models didn’t show its proposed model.

---

## Round 0.2 · Minor Revisions

Authors should add more data and results in the related works section, as well as addressing the reviewers' remaining comments.

Reviewer 1 ·

Basic reporting

See attached

Experimental design

See attached

Validity of the findings

See attached

Additional comments

See attached

Annotated reviews are not available for download in order to protect the identity of reviewers who chose to remain anonymous.

Reviewer 2 ·

Basic reporting

-The authors write the acronyms incorrectly. The correct way is to write them with the first initial letter the meaning of the acronym. such as "Least Absolute Shrinkage and Selection Operator (LASSO)". This error should be fixed throughout the document.
-Authors must avoid the use of apostrophes.
-The authors have not given a space between word and parentheses.
- The terms Figure, Section, Equation, Table, Algorithm, must not be abbreviated and must be written completely.
-In Section 1, it is recommended to divide between the Introduction Section and the Related Works Section.
-At the end of the Introduction Section or Related Works, the authors must include a brief introduction of the Sections that the manuscript will contain.
-Use the acronyms correctly when mentioned in the text as in line 198.
-There is no corresponding space on line 201

Experimental design

- The explanation of the algorithm mentioned in line 208 must be better described and use an algorithm. Authors should be based on https://es.overleaf.com/learn/latex/Algorithms.
-Equation 1 and 2 must explain each of the terms. In addition, the authors must separate the symbol # from all the Equations that they have written in the article.
-On line 271 the URL link must not be placed. So, it must be better cited in a reference.
-The Figures, Tables, Algorithms, must be as close as possible to where they have been cited in the text. It is a terrible presentation for the reader to look for these objects at the end of the article.
-There are too many blank spaces in the manuscript.
-The Figures do not have titles, nor unit in the axes.
-The Figures lack a legend.
-Figure 5 cannot display axis names with "_".
-Authors, when predicting a better classification, must include a classification without the prediction and another with the proposal, in order to see if it predicts or not.
-The authors must demonstrate experimentally and with more details the study they have proposed.

Validity of the findings

-Subsection 322 is too small, authors must include more details.
-The authors have not included future work.
-The authors must improve the conclusions.

Reviewer 3 ·

Basic reporting

See below

Experimental design

See below

Validity of the findings

See below

Additional comments

The paper was incorporated all the comments and also address all the highlighted areas. So, according to my point of view, the paper will be accepted and ready to publish.

---

## Round 0.3 · accepted · Accept

The author have overcome all raised queries, hence I recommend acceptance